# Stable Carbon Isotope Analysis of Hexachlorocyclohexanes by Liquid–Liquid Extraction Gas Chromatography Isotope Ratio Mass Spectrometry: Method Evaluation and Applications

**DOI:** 10.3390/molecules27092874

**Published:** 2022-04-30

**Authors:** Cuiping Gao, Yunlong Wang, Yu Xia, Haixian Liu, Weiguo Cheng, Yi Xie, Yuesuo Yang

**Affiliations:** 1Key Laboratory of Eco-Restoration of Regional Contaminated Environment, Shenyang University, Ministry of Education, Shenyang 110044, China; gaocuiping@syu.edu.cn (C.G.); wyldlnu@163.com (Y.W.); ayaxiayu@foxmail.com (Y.X.); 2China Coal Technology & Engineering Group Shenyang Engineering Company, Shenyang 110015, China; liuhaixian1984@gmail.com; 3Shenyang Academy of Environmental Sciences, Shenyang 110167, China; chengweiguo@syhky.com; 4Liaoning Provincial Ecology & Environment Monitoring Center, Shenyang 110161, China; xieyi_jlu@126.com; 5Key Laboratory of Groundwater Resources and Environment, Jilin University, Ministry of Education, Changchun 130021, China

**Keywords:** hexachlorocyclohexanes (HCHs), compound specific isotope analysis (CSIA), enantiomer specific isotope analysis (ESIA), liquid–liquid extraction, groundwater

## Abstract

Compound specific isotope analysis (CSIA) and enantiomer specific isotope analysis (ESIA) are powerful tools for assessing the fate of hexachlorocyclohexanes (HCHs) in the environment. However, there is no systematic study on the CSIA and ESIA analysis test methods of the carbon isotopes of HCHs in water and soil environments, in particular the isotope fractionation in the pre-concentration process. We endeavored to test the compatibility of CSIA and ESIA with the liquid–liquid extraction method of HCHs in water. The results showed that there were negligible changes in the δ^13^C of HCHs after extraction, indicating that liquid–liquid extraction can be used as a pre-concentration method for the determination of δ^13^C of HCHs in water. The optimized method was validated and then applied to differentiate three HCHs from different manufacturers, to identify in situ degradation of HCHs of groundwater from a contaminated site and to resolve the carbon isotope fractionation occurring in the α-HCH oxidation by CaO_2_/Fe(II) Fenton system. The results showed that the same reagents from different manufacturers have different carbon isotope compositions, and different isomers from the same manufacturer also have different isotope compositions, showing useful evidence in identifying the source of HCHs. The more enriched δ^13^C in the down-gradient wells indicated that HCHs have undergone biodegradation or/and chemical reactions in the groundwater system of the site. Carbon isotopic enrichment factors (ε_C_) of −1.90 ± 0.10‰ were obtained in the oxidation process. Hence, the method validated in this study has great potential as a method for identifying the degradation of HCHs in a water environment.

## 1. Introduction

Hexachlorocyclohexanes (HCHs) have been widely used as pesticides to control agricultural pests since the middle of the 20th century. HCHs have gradually become global pollutants and now pose environmental risks at various levels. In addition to their persistent and toxic bioaccumulation, HCHs pose a high risk to both ecosystems and human health [1]. HCHs (α-, γ-, and β-HCH) were listed in the Stockholm Convention on Persistent Organic Pollutants in 2009 [2]. China is one of the largest producers and users of HCHs in the world and is also most seriously affected by HCHs pollution. Although HCHs were banned from production and utilization in China since 1983, due to their environmental persistence, they can still be detected in the atmosphere, surface water, groundwater and soil [3,4,5,6,7]. Therefore, there is an urgent need to study the fate and environmental behavior of HCHs in these environments. In order to evaluate the enrichment and transformation of these organic pollutants in the water environment, traditional methods were undertaken by analyzing the concentration changes in target substances or identifying their intermediate products. It is difficult to establish an accurate mass balance relationship in the reduction process, because the physical processes, e.g., volatilization, adsorption and migration, can also lead to changes in the concentration of pollutants. Sometimes it is very difficult to detect the intermediate products. In addition, the environmental system is very complex, and the same substance may be from different sources. These problems complicate the assessment of organic pollutant reduction and challenge traditional methods. Compound specific isotope analysis (CSIA) allows an accurate separation of analyses from complex mixtures and the determination of stable isotope ratios at natural abundances. This breakthrough was accomplished by direct coupling of gas chromatography (GC) and isotope ratio mass spectrometers (IRMS) [8]. The CSIA testing was not disturbed by other reactants, products and elements, and has been widely applied in sources identification, biotransformation pathways and degradation mechanisms of organic pollutants [9,10]. Enantiomer specific isotope analysis (ESIA) based upon the CSIA evidence, improves the accuracy in evaluating the sources and conversion processes of the target pollutants in the environment [11]. However, the determination of the stable isotope ratios of the low-concentration organic compounds in environmental matrices has always been one of the main challenges of CSIA applications [1]. The utilization of CSIA for organic matter in low concentrations in groundwater and other matrices has also remained one of the main limiting factors for applications in contaminated sites. How to find effective and feasible sample pre-concentration methods within the sensitivity range of existing instruments has been a key issue for CSIA and ESIA applications using GC-IRMS.

In this study, the liquid–liquid extraction method was selected, which is an effective approach for simple experimental procedures, has good flexibility and is free of cross contamination. Until now, some studies have used CSIA and ESIA to study the degradation mechanism, source identification, and field analyses of HCHs (Table 1) [12,13,14,15,16,17,18,19,20,21,22,23,24,25,26]. Ivdra et al. [1] developed an extraction procedure and a clean-up method for measuring carbon isotope ratios for HCHs, DDT, and chlorinated metabolites in soil samples. However, there was no systematic research on the CISA and ESIA of carbon isotope of HCHs and no reliable pre-concentration process for determining the isotope fractionation in water, especially in groundwater. Therefore, in this study, the EA-IRMS and GC-IRMS methods were used to determine the δ^13^C value of pure products of HCHs (α-, β-, γ-, and δ-HCH), in order to evaluate the accuracy of the GC-IRMS method. The δ^13^C values of the HCHs and α-HCH enantiomers were determined by GC-IRMS to assess the detection limits of CSIA and ESIA testing. The liquid–liquid extraction was successfully utilized as a pre-concentration method for groundwater analysis to evaluate the carbon isotope fractionation and enantiomers of HCHs during extraction. Finally, this method was applied to differentiate three HCHs from different manufacturers, identify the in situ degradation of HCHs of groundwater from the contaminated site, and resolve the carbon isotope fractionation occurring during α-HCH oxidation by the CaO_2_/Fe(II) Fenton system.

## 2. Results and Discussion

### 2.1. Calibration of Isotope Ratios of HCHs Reference Samples

EA-IRMS is a method for obtaining a representative average isotopic signature for a whole sample [27]. Isotopic fractionation is less likely to occur during EA-IRMS than that during GC-IRMS, because fractionation can occur during sample volatilization and burning, as well as during the chromatographic process in GC-IRMS [28]. Thus, it is important to use GC-IRMS to calculate the isotopic ratio of a pure individual standard and compare the result to the given EA-IRMS value [29]. So, the δ^13^C values of pure HCHs were initially measured by EA-IRMS.

The measured δ^13^C values of the reference materials USGS62 and USGS40 were −14.76 ± 0.1‰ and −25.58 ± 0.05‰, respectively. The standard deviation was within the instrumental uncertainty (±0.5‰), indicating that the measured results were reliable. The linear regression between the measured values and the true values was performed and shown in Figure 1. The slope of the regression was 1.072, very close to 1, indicating that the instrument was in good working order and had high accuracy. The δ^13^C values of the pure HCHs determined under this system were shown in Table 2. The δ^13^C values of HCHs directly measured by EA-IRMS were used as the measured value *y*, and the two-point correction was substituted into the regression equation to obtain the corrected δ^13^C values. The δ^13^C values of α-, β-, γ-, and δ-HCH were −25.58 ± 0.02‰, −25.69 ± 0.05‰, −27.68 ± 0.03‰, and −26.74 ± 0.03‰, respectively (Table 2). The standard deviations were between 0.02–0.05‰ (<±0.5‰), indicating the results were reliable and accurate. These δ^13^C values of the pure HCHs were used to evaluate the accuracy of the GC-IRMS system.

### 2.2. Resolution of the HCHs and Determination of Their Carbon Isotope Ratios

The δ^13^C values and signal intensity obtained by CSIA and ESIA for different concentrations of α-, β-, γ-, δ-, (+)α- and (−)α-HCH were shown in Figure 2, according to the method described by Jochmann et al. [30]. The *m*/*z* = 44 ion signal intensity of the HCHs increased with the concentration and showed good linearity in calibration coefficients (R^2^ > 0.99). As the concentration decreased, the *m*/*z* = 44 gradually became positive, and the method detection limits (MDLs) were obtained. The MDLs of α-, β-, γ-, δ-, (+)α- and (−)α-HCH were 181 ± 2, 265 ± 3, 177 ± 2, 176 ± 1, 469 ± 8 and 338 ± 23 mV, respectively (Table 2). This meant that the accuracy of the isotope analysis can be guaranteed in a range better than the MDLs. Therefore, it is necessary to ensure that the signal intensity is higher than its MDLs in order to maintain a good accuracy of measurement of the δ^13^C values.

The δ^13^C values of HCHs measured by EA-IRMS and GC-IRMS were shown in Table 2. The averaged δ^13^C values of the α-, β-, γ-, and δ-HCH references measured by the GC-IRMS system were −26.23 ± 0.25‰, −25.90 ± 0.31‰, −27.93 ± 0.31‰, −26.84 ± 0.2‰, respectively. Comparably those measured by the EA-IRMS system were −25.58 ± 0.02‰, −25.69 ± 0.05‰, −27.68 ± 0.03‰, −26.74 ± 0.03‰, respectively, demonstrating relatively good agreement between these two systems. There are slight differences between the two systems, the reasons for which could be, firstly, that the pure HCHs contain little impurities that possess different carbon isotope compositions with HCHs; secondly, different test systems had different analysis paths and different sample processing; finally, some manual operational factors in the sample preparation process might cause some minor analysis errors. Overall, the results of the two systems agreed well, verifying the accuracy of the HCHs carbon isotope ratios measured by GC-IRMS.

For the evaluation of the potential isotope effects induced by the extraction method, the δ^13^C values from the GC-IRMS analysis were considered as references. The measurement results after extraction were shown in Table 2. The obtained δ^13^C values shifts ranged from 0‰ to 0.21‰, showing no significant difference compared to that before extraction. The results showed that shifts in the δ^13^C values before and after the HCHs extraction from groundwater were in good agreement and a minor difference might be from measurement uncertainties.

### 2.3. Applications

#### 2.3.1. Source Identification

First, 100 mg/L HCHs standard solutions were measured from three different manufacturers from Germany and China. The results showed that the same reagents from different manufacturers have different carbon isotope compositions, and different isomers from the same manufacturer also have different isotope compositions (Table 3), which were different from the results of Chartrand et al. [17]. These differences were affected by the raw materials and/or synthesis process [10,31], which provided data support for the source identification of HCHs in contaminated sites.

#### 2.3.2. Identification Degradation of HCHs in Groundwater from Contaminated Site

The δ^13^C values of HCHs of contaminated groundwater samples were analyzed according to the method described above. The results (Table 4) showed that the distant pollution source (D2) was more enriched in δ^13^C (α-, β-, γ-, and δ-HCH) than the nearby pollution source (D1), indicating that HCHs undergo biodegradation or/and chemical reactions during groundwater migration. Moreover, there was more enrichment in δ^13^C of (−)α-HCH than that of (+)α-HCH, indicating the occurrence of biodegradation.

#### 2.3.3. The α-HCH Oxidation by the CaO_2_/Fe(II) Fenton System

Oxidation by the CaO_2_/Fe(II) Fenton system led to 96.6% degradation of α-HCH within 4 h (Figure 3a). The carbon isotope ratio of α-HCH was enriched in ^13^C from −30.42 ± 0.1‰ to −23.95 ± 0.1‰ (Figure 3b) and gave ε_C_ of −1.9 ± 0.1‰. The ε_C_ was similar to that of the indirect photolysis (UV/H_2_O_2_, ≥280 nm, ε_C_ = −1.9 ± 0.2‰) [15]. Therefore, this may indicate the same mechanism for both systems, demonstrated a dichloroelimination involving two-electron transfers to α-HCH with the cleavage of two C–Cl bonds being expected as the initial step of the reaction. Electrons provided by CaO_2_ may attack HCHs by two-electron transfers eliminating two chlorine atoms from the rings and forming a double bond [15].

## 3. Materials and Methods

### 3.1. Chemicals and Reagents

α-HCH (purity > 98.1%), β-HCH (purity > 98.6%), γ-HCH (purity > 98.8%), and δ-HCH (purity > 99.2%) were purchased from Dr. Ehrenstorfer (Augsburg, Germany). USGS40 and USGS42 were purchased from US Geological Survey (Reston, VA, USA). Analytical grade HCH mixes (α:β:γ:δ = 1:1:1:1) (98.1% purity) were obtained from TMRM (Shanghai, China) and Agro-Environmental Protection Institute, Ministry of Agriculture (Tianjin, China). Hexane and methanol (purity > 99.5%) were obtained from Fisher Scientific (Waltham, MA, USA). Anhydrous sodium sulfate (Na_2_SO_4_, purity ≥ 99.5%), sodium chloride (NaCl, purity > 99.0%), and ferrous sulfate heptahydrate (FeSO_4_∙7H_2_O, purity > 99.0%) were purchased from Tianjin Damao Chemical Reagent Factory (Tianjin, China). Calcium peroxide (CaO_2_, 70.0%) was obtained from Shandong Xiya Chemical Co., Ltd. (Shandong, China). Helium, carbon dioxide, oxygen (purity ≥ 99.999%), and dry compressed air were also prepared. Milli-Q water (resistivity of 18.2 MΩ·cm) was obtained using a Millipore ultra-pure water system.

### 3.2. EA-IRMS Measurement for Determination of Reference Values

EA-IRMS analysis of the pure HCHs standard was performed on a MAT 253 IRMS coupled to a ConFlo IV interface linked to a Flash 2000 EA operated in continuous-flow mode (Thermo Fisher Scientific, Waltham, MA, USA). A sample of 0.14–0.17 mg of pure HCHs standard was wrapped in a tin cup, and four duplicates were analyzed. The oxidation reactor was set to 960 °C and the GC column temperature was set to 50 °C. A two-point calibration was employed vs. the Vienna Pee Dee Belemnite (VPDB) scale, by means of USGS40 (−26.389‰) and USGS62 (−14.79‰) from the US Geological Survey, as described elsewhere [32,33]. The stable carbon isotope ratio ^13^C/^12^C is expressed as δ^13^C in ‰ and the carbon isotope value is expressed according to Equation (1):δ^13^C = *R_sample_*/*R_standard_* − 1,(1)
where *R_sample_* and *R_standard_* were the ratios of ^13^C/^12^C for the sample and standard, respectively. All δ^13^C values were relative to the VPDB standard (δ^13^C_VPDB_) [32].

### 3.3. GC-IRMS Conditions for Carbon Isotope Analysis

The CSIA of the HCHs standard was performed on a MAT 253 IRMS coupled to a GC Trace GC 1310 via a GC Isolink II combustion reactor interface (Thermo Fisher Scientific, Bremen, Germany) operating in continuous-flow mode. The samples were oxidized by passage through a combustion oven containing NiO/CuO operated at 1000 °C. The carrier gas was helium, and the flow rate was 2 mL/min. The injection was in spitless mode, and a ZB-1 GC capillary column (60 m × 0.32 mm × 1 μm, Phenomenex, Aschaffenburg, Germany) was used. The initial temperature was 40 °C and was held for 5 min. The temperature was then increased to 100 °C at a rate of 10 °C/min and was held for 5 min, then increased to 180 °C at a rate of 10 °C/min, then increased to 230 °C at a rate of 3 °C/min, then increased to 260 °C at a rate of 10 °C/min and was held for 10 min. The inlet temperature was 280 °C. The injection volume was 1 μL.

The ESIA of the HCHs standard was performed as the same system with the CSIA but equipped with a γ-DEX™ 120 chiral column (30 m × 0.25 mm × 0.25 μm, Supelco, Bellefonte, PA, USA). The initial temperature was 120 °C and was held for 10 min. Then, the temperature increased to 168 °C at a rate of 1 °C/min, then increased to 220 °C at a rate of 20 °C/min and was held for 3 min. The inlet temperature was 250 °C and the helium was used as carrier gas at the rate of 1.5 mL/min. The injection volume was 1 μL.

Three replicates were measured per sample in order to assess the reproducibility. At the beginning of each analysis, CO_2_ reference gas was introduced into the IRMS setup in five pulses for CSIA and three pulses for ESIA in order to calculate carbon isotope data.

### 3.4. GC-ECD for the HCHs Testing

Concentration testing of the HCHs was performed on a Trace GC 1310 Gas Chromatograph (Thermo Fisher Scientific, Waltham, MA, USA), equipped with DB-1701 column (30 m × 0.25 mm × 0.25 μm, Agilent Technologies, Santa Clara, CA, USA) with the following temperature program: the initial temperature was 40 °C and was held for 5 min, then increased to 100 °C at a rate of 10 °C/min and was held for 5 min, then increased to 180 °C at a rate of 10 °C/min, then increased to 230 °C at a rate of 3 °C/min, then increased to 260 °C at a rate of 10 °C/min and was held for 10 min. The inlet temperature was 280 °C. The injection volume was 1 μL.

### 3.5. Experimental Method

#### 3.5.1. Standard Solution

A total of 5.0 mg of crystalline α-HCH, β-HCH, γ-HCH, and δ-HCH were weighed into silver capsules and loaded into the EA-IRMS system. They were dissolved in n-hexane and diluted to a 10 mL volumetric flask to prepare a 500 mg/L standard solution and were stored in the dark. The standard solutions were diluted with n-hexane into 400, 300, 200, 100, 75, 50, 40, 30, 25, 20, 18, 15 mg/L of HCHs solutions for isotopic analysis.

#### 3.5.2. Sample Pre-Concentration

The water samples were transferred into a 1 L separatory funnel and extracted three times with 30 mL of dichloromethane (shaking 15 min, rotation speed 240 r/min). The three extracts were mixed with anhydrous Na_2_SO_4_ and were dried in a 250 mL pear-shaped flask and evaporated with a rotary evaporator under reduced pressure. When the remaining solvent volume was about 2 mL, it evaporated to near dryness with mild nitrogen at room temperature and was diluted to 1 mL with *n*-hexane for analysis. Sulfuric acid was also required for purification when analyzing the actual groundwater samples.

### 3.6. Instrumental Method Detection Limits

The instrumental MDLs in this study were determined according to the Jochmann dynamic average method [30]. The mean δ^13^C values of the 400, 300, 200 mg/L of the HCHs were determined. An interval of ±0.5% has been set around the calculated mean value. This interval incorporates the total analytical error, including the internal reproducibility on triplicate measurements as well as the accuracy of the measurement based on international standards. As a moving average, the δ^13^C value of the next lower concentration level was added into the calculation of the mean value in the next round. For triplicate measurements, the last concentration whose δ^13^C value was within this interval and for which the standard deviation was less than ±0.5% is defined as the MDLs.

### 3.7. Groundwater Samples

Groundwater samples were collected from a former pesticide plant using a bladder pump (Geocontrol PRO, Geotech, Denver, CO, USA) at low flow (10 mL/min). Each well was purged prior to sampling until dissolved oxygen (DO), pH, electrical conductivity (EC) and oxidation reduction potential (ORP) were stabilized. Two 1 L groundwater were collected in glass bottles and capped with a Teflon septum without headspace, in order to avoid evaporation. All samples were stored at 4 °C until extraction.

### 3.8. Oxidation Experiment

Oxidation experiments were conducted in a batch mode by using a series of 250 mL glass reactors at 25 ± 2 °C. The equilibrated α-HCH solutions were added into the reactor and then diluted to 0.5 mg/L, and then sealed the reactor and mixed the solution with an 800 rpm stirring speed overnight. The experiments showed that there were limited effects in the concentration caused by the volatilization and absorption of α-HCH. The initial solution pH was adjusted to 2 using H_2_SO_4_ solution, and the reaction started after adding the predetermined dose of Fe(II) reagent and CaO_2_. At the desired time, 20 mL samples were transferred to a bottle and immediately mixed with 1 mL methanol to quench the reaction and were analyzed by GC and GC-IRMS after pre-concentration. Each sample was analyzed in triplicate and the average values were displayed.

## 4. Conclusions

The testing accuracy of the δ^13^C values for HCHs measured by GC-IRMS was verified by comparison with those obtained by using EA-IRMS, and a minor difference between two systems verified the accuracy of the HCHs carbon isotope ratio measured by GC-IRMS in our study. We demonstrated that liquid–liquid extraction did not result in isotopic fractionation of HCHs solutions from water samples, supporting that the method can provide accurate δ^13^C values of HCHs from pre-concentration aqueous samples. A difference in δ^13^C between α-, β-, γ-and δ-HCH of different manufacturers was observed, suggesting that CSIA is capable of distinguishing between different HCHs sources. CSIA and ESIA of the real-world groundwater samples from a contaminated site provided evidence that biodegradation and/or chemical reaction of HCHs occurred at the site. Oxidation by the CaO_2_/Fe(II) Fenton system of α-HCH gave ε_C_ of −1.9 ± 0.1‰, indicating that the dichloroelimination involving two-electron transfers to α-HCH with the cleavage of two C-Cl bonds was expected as an initial step of the reaction mechanism. This proposed method thus enables applications of CSIA to contaminated sites for tracking HCHs sources, transformation processes and potential degradation pathways. This method can also provide useful evidence in identifying the chemical and/or biochemical reactive mechanism of HCHs.

## Figures and Tables

**Figure 1 molecules-27-02874-f001:**
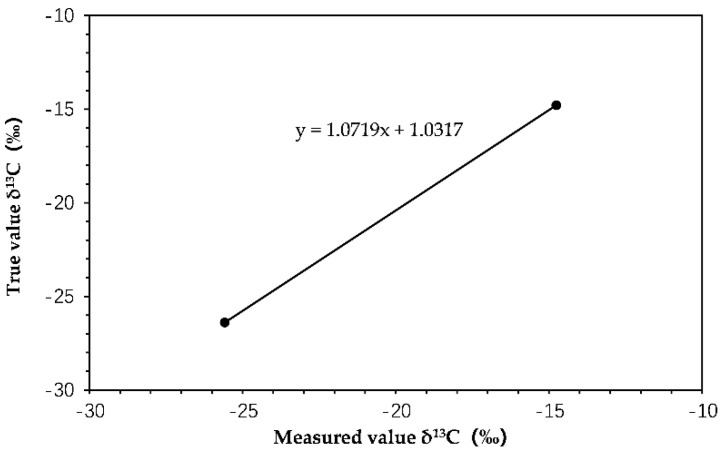
The fitting curve between measured and true values of the isotopic reference materials.

**Figure 2 molecules-27-02874-f002:**
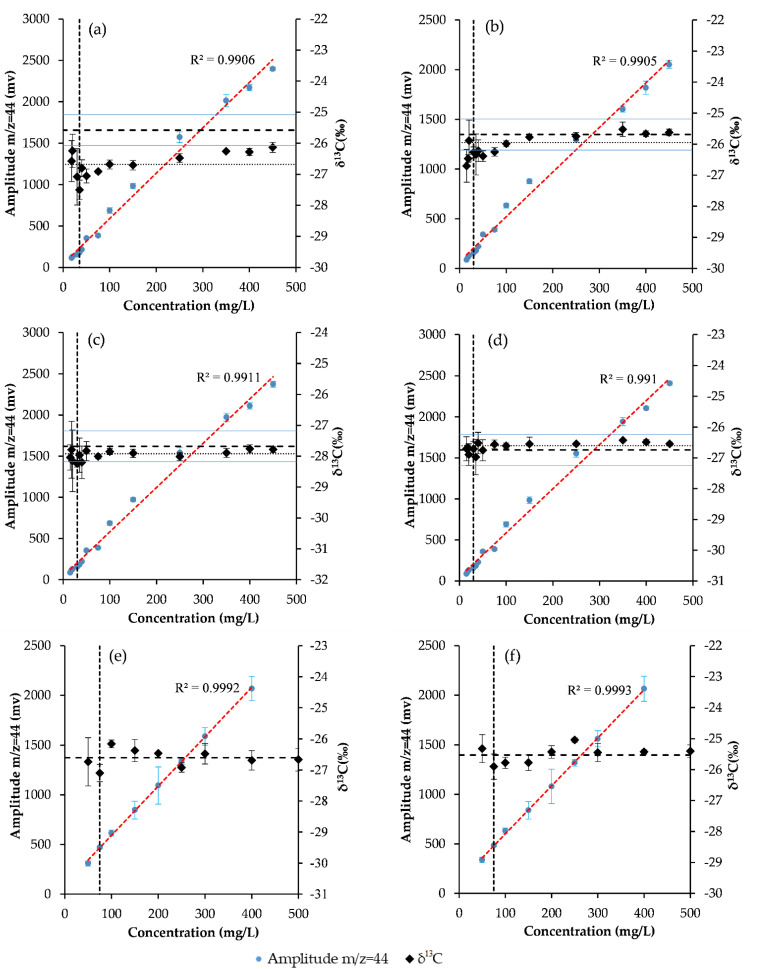
The method detection limits and linearity range of HCHs. (**a**) α-HCH (**b**) β-HCH, (**c**) γ-HCH, (**d**) δ-HCH. (**e**) (+)α-HCH (**f**) (−)α-HCH. The diamonds represent the δ^13^C values in per mil and the squares show the amplitude of mass 44 in mV. The dashed lines represent the intervals of δ^13^C measured by EA-IRMS ± 0.5‰. Each point was measured three times, and the error bars represent the standard deviation.

**Figure 3 molecules-27-02874-f003:**
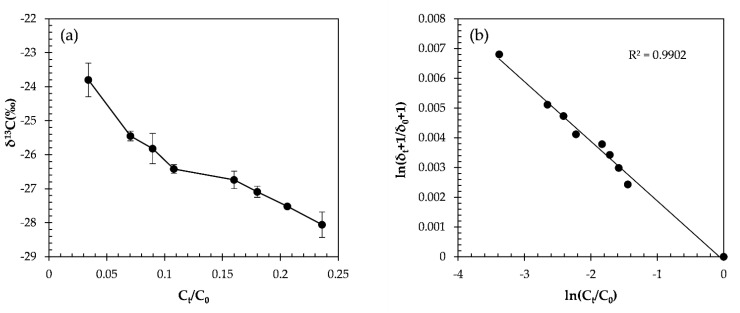
The α-HCH transformation by CaO_2_: (**a**) remaining fraction and carbon isotope ratios for oxidation by CaO_2_; (**b**) double logarithmic plot according to the Rayleigh equation.

**Table 1 molecules-27-02874-t001:** Recent applications of CSIA and ESIA in the HCHs studies.

Methods	Isotope	Scale	Contaminants	Applications	References
CSIA	C	Batch	γ-HCH	Biotic degradation identification	[12]
	C	Field	α-, β-, γ-HCH	Source identification; in situ biodegradation identification	[17]
	C	Field	α-, β-, γ-, δ-HCH	In situ biodegradation quantification	[16]
	C, H, Cl	Batch	α-, β-, γ-, δ-HCH	Source identification	[19]
	C, H	Batch	γ-HCH	Biotic degradation identification	[22]
	C, Cl	Field	α-, β-, γ-, δ-HCH	In situ biodegradation identification	[23]
	C, Cl	Batch	α-HCH	Abiotic degradation identification	[26]
	Cl	Batch, field	α-, β-, γ-, δ-HCH	Source identification; in situ biodegradation identification	[25]
CSIA, ESIA	C	Batch, field	α-HCH	Source identification; in situ biodegradation identification	[13]
	C	Batch	α-HCH	Biotic degradation identification	[14]
	C	Batch	α-HCH	Abiotic degradation identification	[15]
	C	Field	α-, β-, γ-, δ-HCH	In situ biodegradation quantification	[18]
	C, H	Batch	α-, β-, γ-, δ-HCH	Biotic degradation identification	[21]
	C	Batch	α-HCH	Biotic degradation identification	[20]
	C	Batch	α-, γ-HCH	Biotic degradation identification	[24]

**Table 2 molecules-27-02874-t002:** Comparison between EA-IRMS and GC-IRMS for HCHs, α-HCH enantiomers, respectively and the MDLs of HCHs, after liquid–liquid extraction.

Compounds	EA-IRMS/‰ ^2^	GC-IRMS/‰				Amplitude Mass 44 at MDL (mV) ^2^
		Pure Chemicals ^2^	Liquid–Liquid Extraction (mg/L)			
			200 ^2^	300 ^2^	400 ^2^	
α-HCH	−25.58 ± 0.02	−26.23 ± 0.25	−26.05 ± 0.17	−26.21 ± 0.15	−26.19 ± 0.28	181 ± 2
β-HCH	−25.69 ± 0.05	−25.90 ± 0.31	−25.90 ± 0.21	−25.90 ± 0.28	−25.90 ± 0.30	265 ± 3
γ-HCH	−27.68 ± 0.03	−27.93 ± 0.31	−27.98 ± 0.32	−27.73 ± 0.25	−27.69 ± 0.21	177 ± 2
δ-HCH	−26.74 ± 0.03	−26.84 ± 0.25	−26.64 ± 0.22	−26.63 ± 0.15	−26.68 ± 0.18	176 ± 1
(−)α-HCH	n.d.^1^	−26.61 ± 0.39	−26.73 ± 0.46	−26.58 ± 0.39	−26.90 ± 0.05	469 ± 8
(+)α-HCH	n.d.^1^	−25.44 ± 0.46	−25.44 ± 0.33	−25.46 ± 0.14	−25.40 ± 0.13	338 ± 23

^1^ not determined; ^2^
*n* = 3.

**Table 3 molecules-27-02874-t003:** The δ^13^C values of HCHs from different manufacturers.

Sample ID	δ^13^C/‰ (*n* = 3)					
	α-HCH	β-HCH	γ-HCH	δ-HCH	(−)α-HCH	(+)α-HCH
A	−24.94 ± 0.18	−25.61 ± 0.20	−27.08 ± 0.24	−25.62 ± 0.21	−26.61 ± 0.39	−25.44 ± 0.46
B	−24.63 ± 0.18	−25.30 ± 0.09	−26.86 ± 0.10	−29.56 ± 0.17	−26.30 ± 0.19	−25.41 ± 0.22
C	−26.13 ± 0.20	−25.94 ± 0.10	−27.64 ± 0.17	−35.52 ± 0.16	-	-

**Table 4 molecules-27-02874-t004:** The δ^13^C of HCHs of groundwater from contaminated site.

Sample ID	δ^13^C/‰ (*n* = 3)					
	α-HCH	β-HCH	γ-HCH	δ-HCH	(−)α-HCH	(+)α-HCH
D1	−25.63 ± 0.18	−26.30 ± 0.09	−25.86 ± 0.10	−22.56 ± 0.13	−25.61 ± 0.39	−26.74 ± 0.46
D2	−22.34 ± 0.18	−24.51 ± 0.20	−24.05 ± 0.21	−25.74 ± 0.31	−23.30 ± 0.19	−22.36 ± 0.32

## Data Availability

Not applicable.

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
