# Peer review of "Stable Carbon Isotope Analysis of Hexachlorocyclohexanes by Liquid–Liquid Extraction Gas Chromatography Isotope Ratio Mass Spectrometry: Method Evaluation and Applications"

_molecules, 2022, doi:10.3390/molecules27092874_

Round 1

Reviewer 1 Report

The paper describes the development and applications of compound specific isotope analysis to characterize hexachlorinated cyclohexanes in sites compounded with contaminants. The developed method also provides instrument to establish the mechanisms of HCH contamination. The paper overall, is technically well grounded and well written. It can be accepted in its current form.

Per my comment, the paper is technically well grounded. Here are some minor comments.   1. Some paragraphs need to be checked for grammatical errors.   2. Figures 1, 2, Tables 3, 4 have poor resolutions.   3. tables and graphs may be improved by adding colors and indicators.   4. Table 1 does not conform with MDPI formats.   5. The tense in the objectives must be in the past tense.   This will be all.

Author Response

Please see the attachment in PDF.

Author Response

Please see the attachment in PDF.

Round 2

Reviewer 2 Report

Dear Editor,

The manuscript "Stable Carbon Isotope Analysis of Hexachlorocyclohexanes by Liquid-liquid Extraction -Gas Chromatography-isotope Ratio Mass Spectrometry: Method Evaluation and Applications" (molecules-1640509), has been properly improved after revision. I have not any further suggestions. I believe the study is ready to be accepted and published in Molecule Journal.